# Formation of a hard surface layer during drying of a heated porous media

**Navneet Kumar** **\*, Jaywant H. Arakeri, Musuvathi S. Bobji**

Department of Mechanical Engineering, Indian Institute of Science, Bangalore, India

\* navneetkumar@iisc.ac.in, navneet01011987@gmail.com

## Abstract

We report surface hardening or crust formation, like caking, during evaporation when a porous medium was heated from above using IR radiation. These crusts had higher strength than their closest counterparts such as sandcastles and mud-peels which essentially are clusters of a partially wet porous medium. Observed higher strength of the crusts was mostly due to surface tension between the solid particles, which are connected by liquid bridges (connate water). Qualitative (FTIR) and quantitative (TGA) measurements confirmed the presence of trapped water within the crust. Based on the weight measurements, the amount of water trapped in the crusts was ~1.5%; trapped water was also seen as liquid bridges in the SEM images. Further, in the fixed particle sizes case, the crust thickness varied slightly (only 10–20 particle diameters for cases with external heating) while with the natural sand whole porous column was crusted; surprisingly, the crust was also found with the hydrophobic glass beads. Fluorescein dye visualization technique was used to determine the crust thickness. We give a power-law relation between the crust thickness and the incident heat flux for various particle sizes. The strength of the crust decreased drastically with increasing hydrophilic spheres diameter while it increased with higher surface temperature.

**Data Availability Statement:** Data, along with the associated videos, required for reproducing the results of this study are available at, https://figshare.com/articles/Data_-_Crust_related_experiments/7430897.

## Introduction

Evaporation is ubiquitous in nature, from bare water surfaces, soils, and plants, and is useful in many industrial applications. Of major interest is evaporation from the soil, a porous medium, due to its complexities such as a vast range of particle sizes, local textural contrasts etc. As the soil dries, it forms a crust near the surface. The crust formation has been reported to be due to two factors—biological and physical. The latter is observed due to the continuous heating and cooling process, so-called the freeze-thaw action, along with the rainfall effects have been held responsible for the crusting of the upper layers. Thus along with leaching, rainfall seems equally important in the formation of physical soil crusts. The hardening of soils is thus a common observation in nature whether it is in the form of mud-peels or soil crusts. Mudcracks followed by peeling off of a thin soil layer (mud-peeling) is a common observation when a water source (such as a pond, lake, or a river) runs dry for a long time. Scientists have long puzzled over the formation of these cracks while the phenomenon of mud-peeling has hardly been explored. Numerous experiments successfully imitated cracking in the laboratory [1–8] and in

**Funding:** JHA received funding from Robert Bosch Centre for Cyber-Physical Systems under the grant RBCCPS/ME/JHA/PC-0013. MSB was the co-investigator. The funders had no role in study design, data collection and analysis, decision to publish, or preparation of the manuscript.

**Competing interests:** The authors have declared that no competing interests exist.

the field [9,10] but only a few of them observed the mud-peels [1,6]. These experiments were conducted with river bed sand [1], starch solution [3], coffee water mixture [2], suspension of latex particles [5], cement [9,10], and concretes [7]. Peeling-off (followed by cracking) of colloidal suspensions [11–17] such as wet paints during and after its drying (once it has been applied on a surface) is another commonly observed phenomenon. It has been shown that the cracking mechanism is different for soft and hard particles. Evaporation leads to compressive capillary forces on the particles [11,18,19] and in case of hard particles, cracking is a pressure release mechanism [9,13]. Mud-peels (S1C Fig in S1 File), in general, are flakes of a thin layer of particles adhered (via traces of water or salt or some other chemical reaction based mechanisms) together providing it with some strength compared to the layer just below it. 'Leaching', loss of nutrients in the form of minerals and salts, brings salt to the top and deposit them in a few top (exposed to the ambient) layers of soils making it hard [20–22]. It has been known for quite some time that the addition of a small amount of water changes (increases) the soil strength significantly [23]. On the other hand, fully saturated and fully unsaturated soils have been found not to resist the shear force and hence could be treated as having no strength at all.

Evaporation from a confined porous medium [24,25] has been shown to undergo three different stages; these stages also existed for other porous systems such as texturally layered [26], a stack of rods [27], and a stack of plates [28]. Recently, with a horizontal stack of rods, it was shown [29] that the rods' surface roughness is important in deciding the duration of these stages. In the 1$^{st}$ stage, a porous medium sustains higher evaporation rate (close to that from a bare water surface) which has been shown to be due to the presence of water [25,26,28] near the porous medium top surface. In stage 2 of evaporation, the evaporation rate reduces drastically to very small values; the transition stage connects stages 1 and 2. The surface water content is zero in stage 2 and evaporation occurs far from the porous medium top surface. Note that, in these type of processes, the competition is between the gravitational and interfacial forces while the viscous effects have been suggested [24,30,31] to be taken into account either when the evaporation rates are very high or when the particle sizes are large or both. Water on the porous medium top surface is connected to water in the deeper regions of the porous medium through capillary films [32–34]. Regions where water is drained, by the capillary films, trap some water in the form of pendular structures [23]; co-existence of all the three phases. The (tiny) trapped isolated water exerts enough interfacial force on the solid particles to hold them together [35–37] and is believed to be the main reason behind the formation of sandcastles [38–43] (S1A and S1B Fig in S1 File). This property of unsaturated soil has been used to create sharp-cornered structures which would be otherwise impossible if the soil is either fully wet or fully dry [39]. Using fluorescein microscopy, it was shown [40–42] that liquid bridges form at some critical water volume percentage (calculated as the volume of water present or added to the total sample volume) before which water resides within the roughness of the solid spheres. This critical value varied between different studies (it was 0.07% in case of glass beads with an average diameter of $375 \mu m$ [41]) largely due to variation in bead sizes and packing fractions. At some instant (0.2% in [41]) these liquid bridges were fully developed. The number of these bridges per bead increased with increasing water volume and eventually saturated at ~6.5 at 0.8% water content; these data were reported [40,41] for the case of random close packing where void fraction was ~36%. Further addition of water led to decreasing numbers of bridges [42] as the structure now shifts from pendular to a funicular one [23,43]. The measured tensile strength [42] also increased with increasing addition of water and reached a maximum at 0.015% of water after which it remained constant till 15% of water. A value of 15% of water means that ~35% of available space has been occupied by water; in these experiments, the reported packing fraction was ~0.57. For a perfectly wetting liquid bridge, the interfacial force (*F*), see S11 Fig in S1 File, after the critical water content, remains a constant

[35,37,43] and is given as:

$$F = 2\pi R\sigma \tag{1}$$

Where, $R$ is the grain (solid sphere) radius and $\sigma$ is the interfacial surface tension. It so happens that an increase in the liquid volume is adjusted by a corresponding decrease in the radius of curvature.

Previous investigators added liquids (such as water), in a controlled amount, to spheres and measured the soil strength (either shear or tensile) with or without vertical agitation. Their analysis was mostly focussed on the angle of repose and, in particular, finding the critical angle after which the system slumps. Investigators have also studied this 'sticky' nature of partially wet sand during drying [1,7,20,24,33] but none have reported results on the impact of either particle size or the surface temperature. The formation and 1-D growth of soil crust (near the air interface), during the drying of a colloidal suspension, has been modelled (a moving boundary problem) recently [44] where a constant evaporation rate was assumed. Peeling-off of the upper crusted layer is initiated by horizontal cracks which propagate parallel to the drying surface. Note that a horizontal crack forms only if vertical (mudcracks) cracks are available. It was shown [45] that the gradient in the tensile stress at the drying surface forces the crack to curl up and form a mud peel; similar to 'spalling' and 'caking'. The depth of peeling was also modelled [45] considering a constant water evaporation rate at the drying surface. Apart from a few of these studies, there are no concrete evidence behind the formation and strength of the crusts in a drying porous medium.

We report laboratory experiments on the formation of crusts (during drying) in porous media consisting mostly of nearly mono-disperse glass beads; similar-sized spheres have been used previously [33,40,42,43]. We have defined 'crusts' in our experiments as 'the hard layer formed, at the side exposed to the ambient, during the drying of a porous medium'. Majority of the experiments were conducted, with de-ionized (DI) water, while heating the samples from above using infrared (IR) radiation. We found that a thin upper (exposed to the open boundary) layer of the sample was crusted (neither very flaky nor like a slump) which fragments into pieces (like a cookie) when broken; this has never been reported previously. We attempt to answer a few fundamental questions like (1) when and where do crusts form, (2) why do they form at all, and (3) how do they form? We further investigate the dependence of the crust properties (such as its thickness and strength) on various controlled parameters such as the surface temperature (or the heat flux incident on the porous medium top surface), particles sizes, and wetting characteristics of the porous medium. A few experiments were also carried out with sand (with a range of particle sizes) and other evaporating liquids.

## Materials and methods

Confined and saturated porous mixtures consisting (mostly) of DI water and glass beads (95% spherical; Spectrum Marketing, Mumbai, India) were prepared following a specific protocol [25,46]. Three different diameters viz. nearly mono-disperse 0.10–0.16 mm, 0.40–0.50 mm, and 0.70–0.85 mm, of glass beads (GB) were used; the average sizes can be considered as 0.13, 0.45, and 0.78 mm in the same order. These glass beads are solid, non-porous, and hard; they don't deform due to interfacial forces. In some experiments, the glass beads were cleaned using piranha solution and in one case sieved natural sand (0.30–0.50 mm particle diameter) was also used. Different evaporating liquids were used such as distilled water, millipore (multi-stage distilled) water, and (analytical grade) acetone. Experiments were conducted in different containers, depending on the duration of stage 1, for different purposes. The container containing the porous medium was insulated from all the sides except at the top. The medium

saturated with water was heated radiatively using a 20cm x 20cm flat ceramic IR heater (Care Systems and Control, Bangalore, India) from the top. The heater was connected to a variac for controlling the IR intensity which was also controlled by varying its distance from the porous medium. Mass loss was monitored using a precision weighing balance (Sartorius GPA5202 with a least count of 0.01g); mass was recorded on a computer every 15 seconds.

The ambient temperature was measured using a T-type thermocouple (Omega Engineering, UK). A Honeywell (USA) humidity sensor (HIH-4000 with an accuracy of 2%) measured the relative humidity (RH) in the ambient away from the heating area. A data logger (Agilent Technologies, USA; Model 34972A) was used to log the temperature from the thermocouples and RH sensor. A schematic of the experimental setup is shown in S2 Fig in S1 File. A thermal camera (Fluke Corporation, USA; Model Ti400, 320x240 pixels) was used to monitor the porous medium top surface temperature at different times. Small and large scale experiments are referred to those types of porous media whose heights were much lesser and larger than the capillary characteristic length [25], respectively. Experiments were also conducted with hydrophobic glass beads; a standard technique [47] was used to make glass beads hydrophobic. Results on these beads are seen in S10 Fig in S1 File while the sample preparation method is detailed in the S1 File.

## Making glass beads hydrophobic

The glass spheres were first cleaned by treating them with the piranha (3:1 $H_2SO_4$ and 30% concentrated $H_2O_2$ mixture) solution. The cleaned glass spheres were rinsed with the distilled water multiple times and were left to dry. The dried clean glass spheres were then poured in a mixture of isooctane and FOTS (fluoroctatrichlorosilane; Sigma-Aldrich, USA) solution; this solution was prepared by mixing 1mM of FOTS with isooctane. The top portion of the container was closed with the wax tissue and was put in an ultrasonicator. After 30 minutes of the treatment, the glass spheres were taken out of the container and were put on a pre-cleaned glass plate. This glass plate was then kept in an oven where the wet glass spheres were heated at 70˚C for 6 hours. The evaporation of the silane leaves a thin film around the glass spheres making them hydrophobic. The glass beads were turned once to get the hydrophobic layer uniformly around them.

## Sample preparation with hydrophobic glass beads

The porous medium was created by mixing the prepared hydrophobic glass spheres with the DI water. An experiment was conducted in a glass beaker with a diameter of 4.6 cm. The sample was ~5cm high. Heat received by the porous medium top surface was ~2000 W/m$^2$; this value was estimated based on the steady state heat losses from the top surface and into the ambient. In this case, the water level was always kept higher (than the top-most glass beads level) which prevents the hydrophobic spheres from popping up to the water surface. Care was also taken while dropping the hydrophobic spheres in water so that no air was entrained along with them in the liquid. The experiment was stopped when no significant mass loss was observed. Surprisingly, the porous medium was found crusted throughout the height unlike with the hydrophilic ones where the crust was limited to a few layers near the top exposed end.

## Treatment of natural sand

Apart from the nearly-monodisperse particle sizes, an experiment was also conducted with sieved natural sand, whose particle diameter ranged from 0.30 to 0.50 mm. Before its use, the sieved sand was oven-dried at 350˚C for two days in order to remove the unwanted constituents. Upon heating, it was slowly cooled, to the room temperature, in order to avoid any

condensation. The sample height (of the porous medium consisting of this sand) was approximately 8 cm with an overall porosity of ~43%.

## Imaging

Three types of microscopes were used for imaging purposes. Larger glass beads were imaged using low magnification (maximum 10x) microscope (Lawrence & Mayo, India). Samples with smaller glass beads were viewed using a high-resolution microscope (Carl Zeiss AG, Germany); the magnifications used were 100x and 200x. In the third type, we used a scanning electron microscope for further magnification and better contrast. Along with these three standard types of microscopy, we also used digital cameras (Nikon P7800 and Canon A2600) for imaging purposes.

## Dye visualization

We used a unique visualization method for tracking the evaporation sites. The benefits of this new type of method, using fluorescein particles as a dye, has been explored recently [25–27,46]. The particles are originally orange in colour but turn water green when mixed with it. Conversely, a fluorescein dye mixed water solution will slowly turn orange upon evaporation. Thus, the solid regions, where water has fully evaporated, are left with the deposited fluorescein particles. Interestingly, the regions from where water has been drained (either by gravity or by capillary films) would finally correspond to the original colour of the solid particles used in preparing the porous medium; in our experiments, it is white, the true colour of glass beads.

## Results and discussion

### Characteristic curves

We briefly discuss the process of drying from a typical experiment. Fig 1 shows the variation of the rate of evaporation, of water, from an initially fully saturated (when all the voids in the porous medium were occupied by the liquid) porous medium consisting of 0.78 mm diameter glass beads. The heat flux received by the porous medium top surface was ~1000 W/m$^2$. In the presence of this heat, the evaporation rate increases rapidly and reaches ~20 mm/day (at an ambient relative humidity of ~60%) within 3 hours; this value is about the same as that from bare water surfaces and is known as the 'potential' evaporation rate. High evaporation rates are sustained in stage 1 (see Fig 1) even as the porous medium dries. The high rates of drying are maintained by the capillary films which maintain the continuity between the water near the exposed end and continuously receding (within the porous medium) drying front. The presence of distinct wet patches was clearly seen using an IR camera [25,26,28] as seen in Fig 2. With time these wet patches shrink (see Fig 2) and eventually vanishes from the exposed surface. At this instant, the porous medium enters stage 2 of evaporation. For the glass beads with 0.78 mm diameter, stage 1 is sustained (see Fig 1) till ~23 hours. The rate of drying drastically decreases in the transition regime (at the end of stage 1) and eventually takes a much lower value in stage 2. Crust forms during both stage 1 of evaporation and in the transition regime. Similar curves of the evaporation rate for a few other porous media are seen in the inset of Fig 1. Apart from the evaporation rate (Fig 1), the other evaporation characteristics curves (temporal variation of mass loss due to evaporation and the near-surface temperature) are seen in S4-S6 Figs in S1 File. A reduction in the evaporation rate is followed by an increase in the surface temperature. In stage 2 of evaporation, the surface temperatures are seen much higher than their values in stage 1; this depends on the amount of heat intercepted at the porous medium top surface.

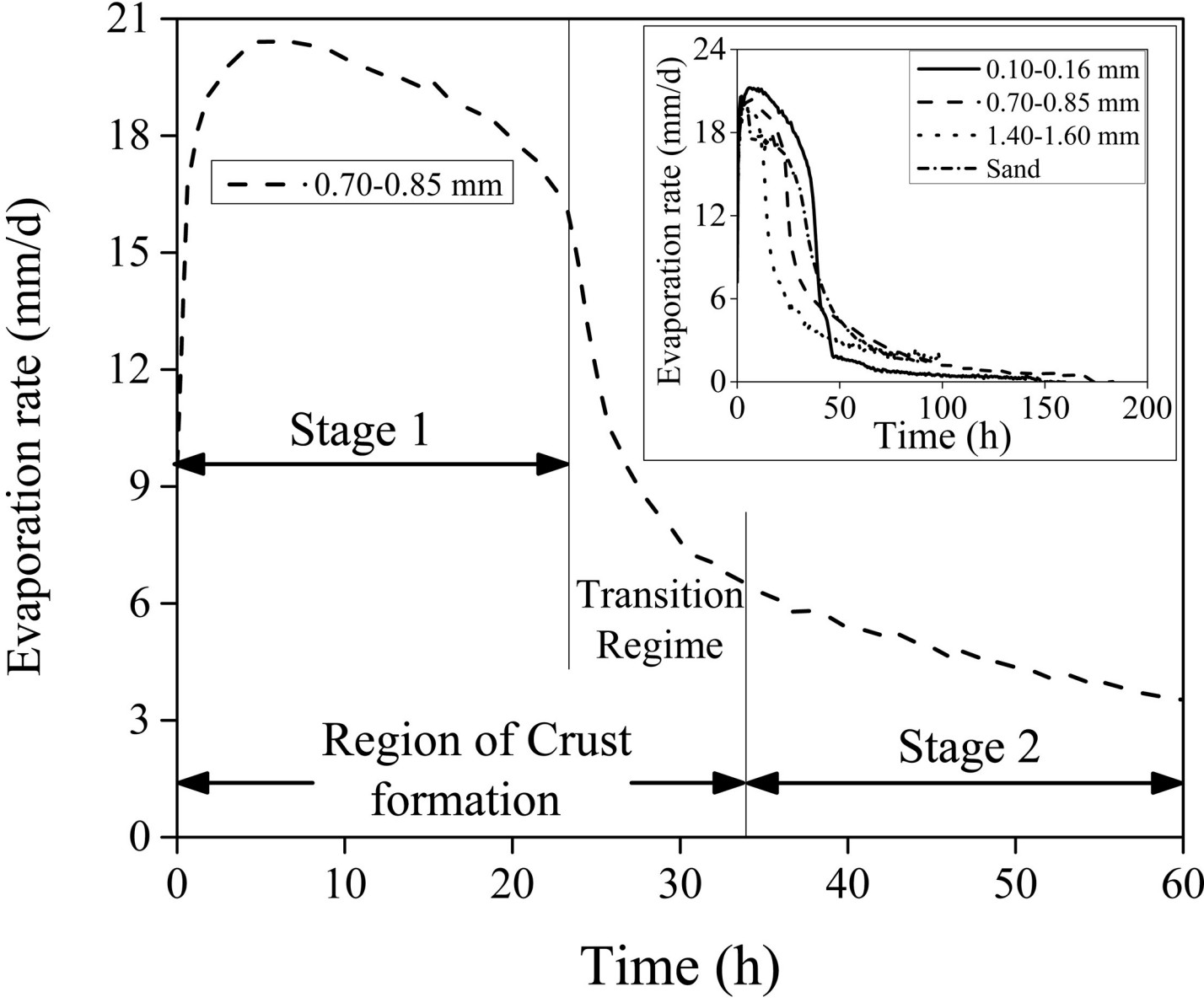

**Fig 1. Variation of the evaporation rate versus time for the (hydrophilic) glass beads with 0.78 mm diameter.** Curves for the other cases are seen in the inset. Heat flux received by the top surface in all the cases was ~1000 W/m². 'Sand' was slightly more porous (~43% porosity) compared to the other cases with the (hydrophilic) glass beads (35–37% porosity).

Fig 2 shows the variation of the evaporation rate, from 0.78 mm diameter glass beads, versus the percentage of remaining water or saturation (S). The curve should be seen *from right to left*. At S = 1, the porous medium is fully saturated; the instant where the experiment began. With time, the porous medium loses water and S decreases. The IR images were taken from the above, of the porous medium top surface, are also seen in Fig 2 at four distinct instances. For clarity, these IR images have been marked with the corresponding S values, the experimental duration, and the (dynamic) temperature scale at their right side. White lines in these images separate the 'wet' and 'dry' regions [25,26,28]. Note that the region within the white line reduces with time (decreasing saturation values) and eventually vanishes at the end of the transition regime (this marks the beginning of stage 2 of evaporation); this has been recently

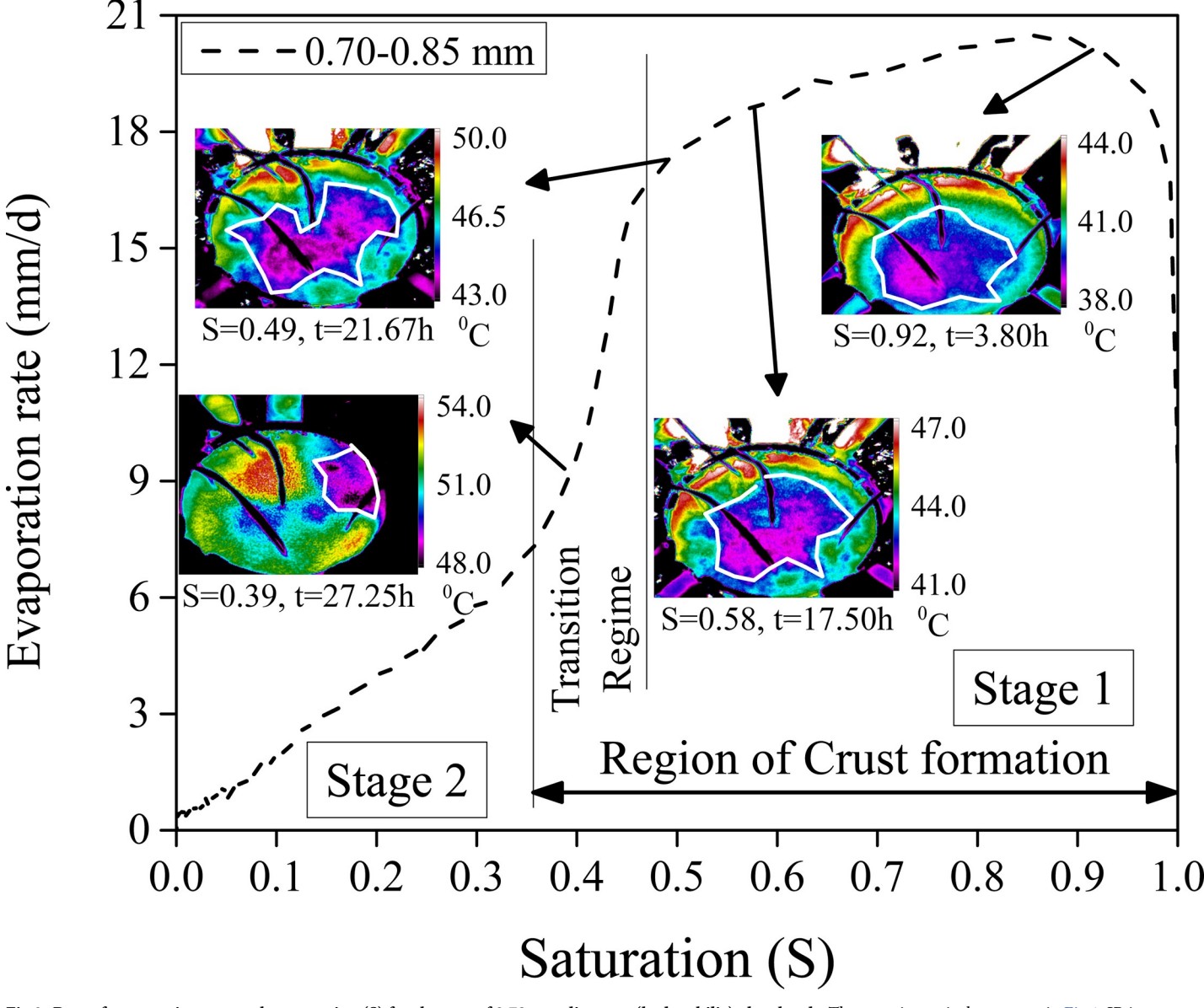

**Fig 2. Rate of evaporation versus the saturation (S) for the case of 0.78 mm diameter (hydrophilic) glass beads.** The experiment is the same as in Fig 1. IR images corresponding to four important instants are also seen. White curved lines in the IR images represent the boundary between the completely wet (inner) and the completely dry regions on the surface of the porous medium. Also mentioned are the temperature scales for the IR images.

named '*shrinking wet patch*' pattern [25]. In connection with the formation of the hardened surface layer, we observed that the region outside the white line was already crusted (even at S value as high as 0.90) while the region within the white line was soft (even at S as low as 0.40). We conclude that the low temperature region in the IR images represent the relatively wetter zone which is soft while the higher temperature regions are hard. The definition and determination of either hard or soft layer are discussed next. Note that the crusts (hard layer) start forming as soon as the water started moving from the relatively larger pores to the smaller ones. We now discuss the formation of crusts (based on the direct experimental observations) for the various cases and we also present a simple (visual-based) method to determine the

crust's thickness. During the passage, we also explain, at the pore scale, where the crusts form and the reason for them.

### Drying in the presence of external IR heating

We now present experimental evidence of the crust formed near the evaporating end; these crusts are only a few particle diameters thick. Crusted surfaces were clearly observed in all the cases with external heating. Crusts were not observed when the samples were not externally heated. Fig 3 shows the condition at different locations of porous media at the end of the experiment. These experiments were carried out in a 6.3 cm diameter glass beaker. The water-saturated sample heights in all the cases were between 8–9 cm and DI water was used as the evaporating liquid.

Heat flux received by the porous medium top surface was ~1000 W/m$^2$ in all the cases. The average surface temperature in stage 1 (wet patch period [25]) was between 36˚C and 42˚C

(a) Hardened thin crust broken in pieces
0.13 mm glass beads

(b) Hard crust sticking to the wall
0.78 mm glass beads

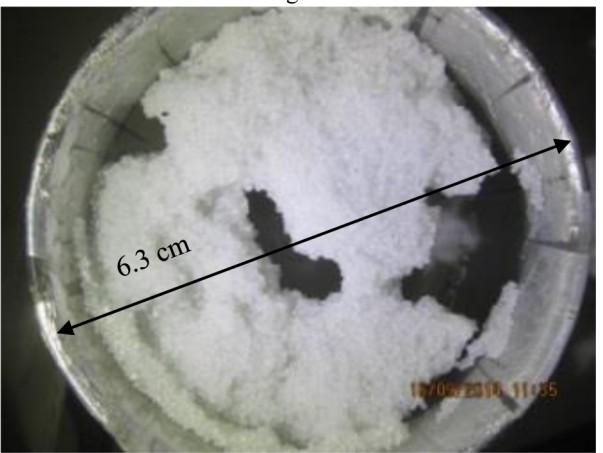

(c) Sandcastle-like 'clumps'; 0.13 mm glass beads
Found after removing the top hard crust layer

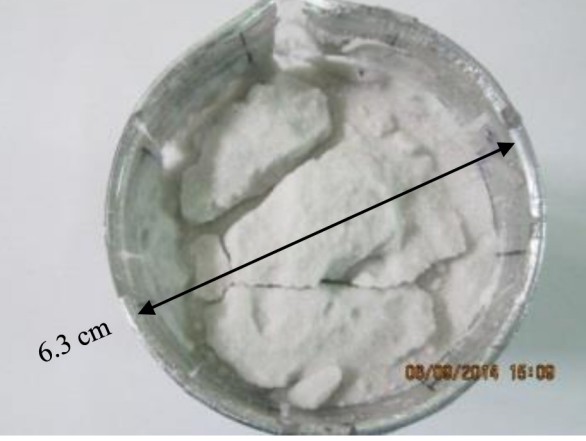

(d) Entire hardened 'sand' column

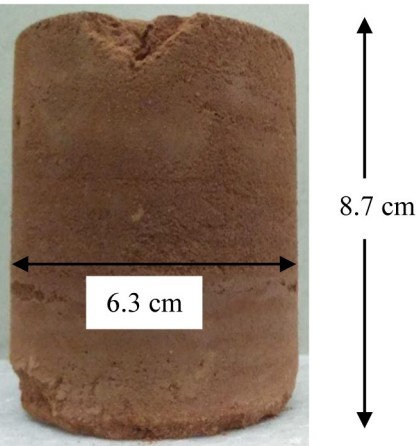

**Fig 3. Images at the end of the experiments with different (hydrophilic) samples.** Crusts, formed within a few top layers, in the case of (a) 0.13 mm and (b) 0.78 mm diameter glass spheres respectively. Clumps, similar to sandcastles, rather than the crusts are seen (c) at the deeper locations away from the top exposed surface. The crust in case of the natural sand (d) is not limited to a few layers near the top but covers the entire column height. Heat flux incident on the porous media top surface in all the above experiments was ~1000 W/m$^2$.

while in stage 2 (dry surface) it was close to 60˚C (see S5 Fig in S1 File). The experiment was concluded either when the porous medium stops evaporating or when the evaporation rate reduces to a very low value i.e. 1–2 mm/d.

Fig 3 shows the conditions of the samples at the end of the respective experiments. At this instant, the porous media were dry except for the tiny liquid bridges between the particles where water was trapped in the form of the pendular structures. Fig 3A shows the thin-crusted sample, with 0.13 mm diameter glass beads, broken in multiple parts; they broke like a cookie. Initially, the entire top surface was hard which was removed carefully. This larger crust (~6 cm diameter and ~2 mm thickness) did not break when held (with fingers on the top and at the back near the periphery) horizontally or vertically. Clusters of spheres were also held by the liquid bridges at deeper regions (away from the top surface) but, unlike the crusts, they are seen as 'clumps' (Fig 3C) similar to the sandcastles. The crusted pieces (Fig 3A) however were much harder than the clumps (Fig 3C). Obviously, unlike the crusts, these clumps fell apart easily. Similar to the 0.13 mm diameter GB case, crusts were also found (see Fig 3B) with 0.78 mm diameter glass beads. An interesting point to note here is the sticky nature of the crust with the container wall. After the end of the experiment, a few holes were made on the top crusted layer. Apart from the spheres present in the crust (~4 mm thick), the remaining spheres below the crust (~85 mm high) were removed, after tilting the container, through the holes in the crust. The image (Fig 3B) is a result of following this procedure. Note that in both glass bead cases the crust was very thin and in fact was limited to a few numbers of layers at the evaporating end. Surprisingly, with the natural sand, the crust was not limited to a few layers but occupied the entire column height (Fig 3D). We believe that this unique feature occurs due to two reasons: (a) presence of multiple particle sizes and (b) irregular geometry of the sand particles. These two reasons enhance the contact area between the particles, which must have provided additional strength for the crust to span the entire porous column. The more detailed information and discussion regarding the results of the experiment with the natural sand are given in the supplementary information (S7B and S7C Fig in S1 File).

The microscopic images seen in Fig 4 present a better picture of the crust (and its strength). Fig 4A, taken at 200x magnification clearly shows the isolated water (which may also contain some impurities) trapped, acting as a liquid bridge, between two 0.13 mm diameter glass beads; these water bridges are marked '1' and '2' in the image. The water bridge is not seen when the spheres were separated by some distance; this gap is ~10–15$\mu m$. This clearly means that during the evaporation process, a water bridge can only be trapped if two spheres are in contact at the level of surface roughness. Note that water is also trapped between the beads and the container wall (not shown here). Similar liquid bridges between spheres in contact are seen (Fig 4B) for the case with 0.78 mm diameter glass beads; this image was taken at 8x magnification. Water bridges were also observed in the case of 1.40–1.60 mm diameter glass beads (S7A Fig in S1 File). The SEM images, taken of the crust with 0.13 mm diameter glass beads, shows a clear and detailed picture of the liquid bridges (Fig 4C and 4D) and the condition of the beads. Some impurities are also seen deposited on the glass beads' surface (and possibly in the water bridges); these impurities may have come either from the sample or from the atmospheric air or both. Fig 4E shows the microscopic image of the dry and unused 0.13 mm diameter glass beads. As expected, liquid bridges were not seen in this case.

Next, we look at the process of the formation of the unevaporated liquid bridge (which was found in the crusted samples discussed earlier). For this, we devised a controlled system consisting of a single liquid bridge trapped between two 3.00 mm diameter glass beads in contact. DI water was coloured green using the fluorescein dye for better visual contrast. The beads were constrained which restricted their motion. It is important to understand the evaporation from such a basic system since this is what must be happening, at multiple locations, in the

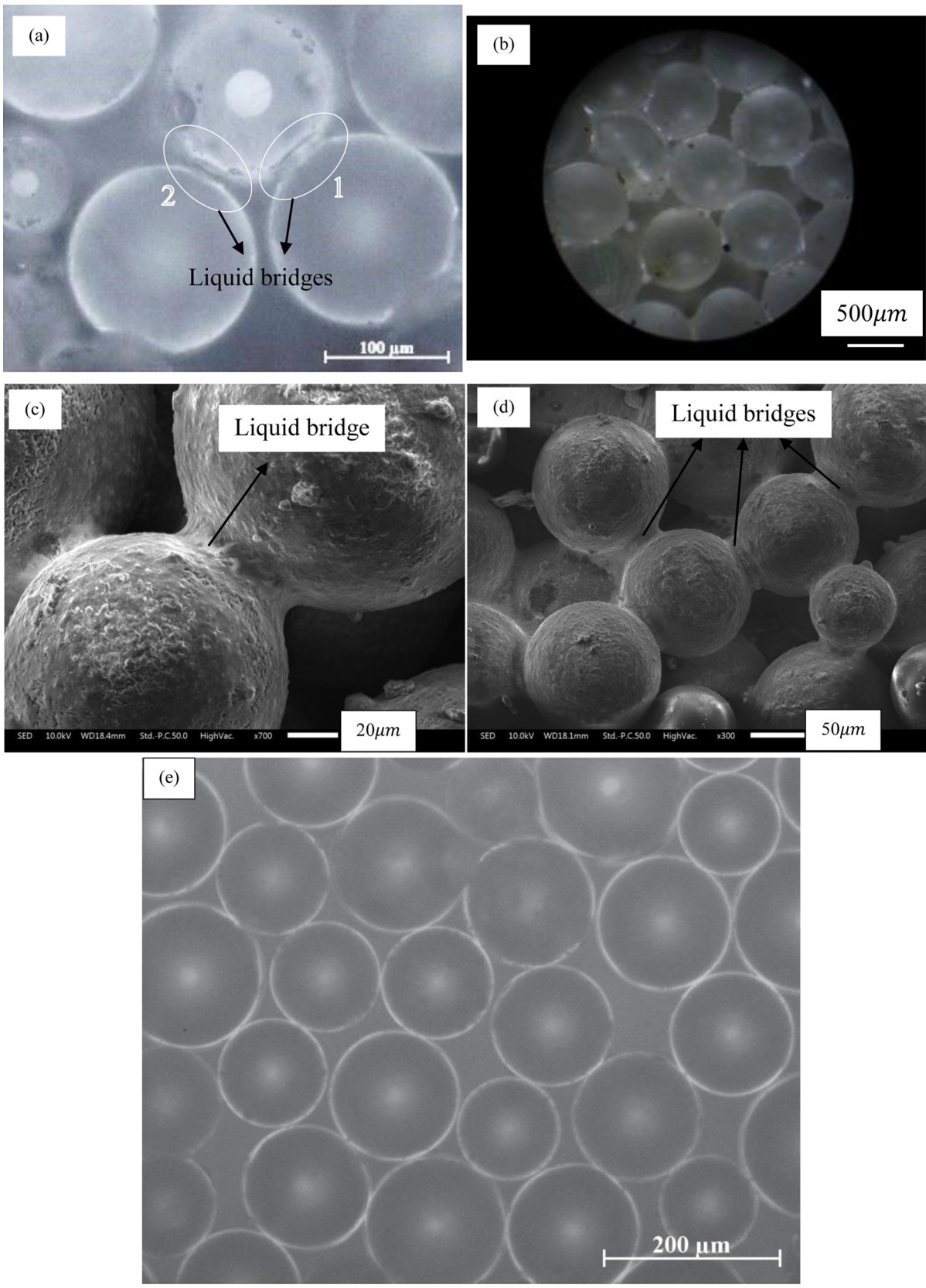

**Fig 4.** Microscopic images of the crust showing (a) two liquid bridges between 0.13 mm diameter glass beads– 100x magnification and (b) liquid bridges at multiple locations between 0.78 mm diameter (hydrophilic) glass beads– 10x magnification. Traces of water in between the glass beads are clearly seen in these images. Under low pressure conditions in SEM, these liquid bridges either led to solidification or particle deposition as seen in (c) and (d); particles used in these images are 0.13 mm diameter (hydrophilic) glass beads. The control image (showing no liquid bridges) of the dry 0.13 mm diameter glass beads are seen in (e).

large scale experiments. The microscopic images (S9 Fig in S1 File) shows, with time, the shrinking liquid bridge and (eventually) a thin unevaporated region which did not evaporate even after many days. A video (S4 Video) shows the drying of this trapped meniscus up to the unevaporated part. A similar phenomenon is expected to occur in the large scale experiments i.e. the contacts between the beads indeed retain some water which remains unevaporated. The reason for the unevaporated water content needs further investigation. It is clear that the presence of water is one of the major reasons behind the crust formation and its strength similar to the ones observed in the sandcastles and mud-peels. An important difference between these two (crust in the present study and sandcastles) systems is the route they take towards stability. In sandcastles, water is added while in the crusts trapped water is a result of evaporation.

We now discuss the rough estimates of the amount of water trapped in different zones at the end of the experiment. In the experiment with the 0.13 mm diameter glass beads, the crust contained ~1.5% (by weight) of water while the clumps retained only about 0.1% (by weight) of water; a total of ~0.5g of water in this experiment did not evaporate even after seven days of heating (see evaporated mass versus time curve for this experiment in S4 Fig in S1 File). The weight percentages of water left in different zones of a porous medium are discussed in detail in the quantitative analysis section. The weight percentage of trapped water in the crust was measured using a precision weighing scale. For the clumps, its weight percentage was calculated based on the total water trapped, water in the crust, and height of the sample. We report, in brief, observations from a few other intuitive experiments, which are directly related to the strength of the crusts. These are:

1. The crust was weaker for the larger bead sizes. With 2.50–3.00 mm diameter glass beads, the crust did not even form.

2. A separate experiment was conducted where the glass beads and the glass beaker were cleaned using the piranha solution and millipore water was used as the evaporating liquid. Crust (weak) formed in this case was due to the water trapped between the beads; though trapped water was not seen clearly (S8A Fig in S1 File).

3. Crust formed (with external heating) with acetone (S8B Fig in S1 File) was much weaker compared to that with water. The interfacial surface tension is therefore crucial in determining the crusts' strength.

4. We didn't notice any significant evidence for the mudcracks in the experiments with the glass beads. Nearly mono-disperse particle sizes and a relatively lower porosity value (36.5 ±1%) in the experiments may have drastically reduced the particle's motion during the evaporation process thereby avoiding the mudcracks.

We have shown, till now, that crusts (hardened layer) are formed during the drying of various porous media heated from above. Unheated samples did not produce any significant crusts. We have also shown that the crusts form due to the unevaporated liquid trapped between the contacts of the spheres. Crusts are harder in case of fluids with higher surface tension. The crust is much stronger if the porous medium consisted of a range of irregularly shaped particle sizes such as sand and soil.

## Water content in the crust

We have established, using the microscopic, SEM images, and mass of the measured crusted samples, the fact that the crusts' strength is due to the water trapped between the particles. Henceforth an attempt was made to precisely quantify the trapped water content qualitatively and quantitatively.

**a Qualitative analysis.** In order to further strengthen the claims, a qualitative approach was taken. Fourier Transform Infrared Spectroscopy (FTIR), a standard technique, which is used to determine the types of bonds in any samples, was used to obtain the absorbed infrared spectrum of the crusted sample; obtained with heating the porous medium. The wavenumber range spanned by FTIR was from 4000$cm^{-1}$ to 400$cm^{-1}$ and the corresponding wavelength is between 2.5μm to 25μm. The Transmittance of the crusted sample is plotted against the wavenumber as seen in Fig 5A. Generally, a FTIR spectrum is divided into two halves around 2000$cm^{-1}$; our interest lies in peaks at wavenumber larger than 2000$cm^{-1}$. Four distinct transmittance peaks are seen corresponding to wavenumbers of 2360, 2831, 2943, and 3315$cm^{-1}$ respectively. These peaks were also found with other crust samples consistently. The peak corresponding to 2360$cm^{-1}$, a weak peak, represents the presence of asymmetrical stretched $CO_2$ bond. Peaks of 2831 and 2943$cm^{-1}$ correspond to the presence of a stretched–C-H bond in the crusted sample. Of more importance is the strong peak at 3315$cm^{-1}$ which clearly indicates the presence of water in the crusted sample. In FTIR spectrum a strong peak close to 3300$cm^{-1}$ [48] also represent a few other functional groups such as stretched ≡C-H, -OH in alcohols and carboxylic acids, the existence of whose is impossible in the present experiments. FTIR spectrum was also produced (Fig 5B) for a dry sample. Obviously, the peak representing water corresponding to 3300$cm^{-1}$ wavenumber is missing in this case.

**b Quantitative analysis.** Curiosity arises regarding the amount (or volume) of water trapped in the crusted samples. We put the crusted samples in an oven at 250˚C for 1 day. No physical change in the crust was observed and the particles were still sticking to one another. We performed thermogravimetric analysis (TGA) of the crusted samples where both the mass loss and the chamber temperature were simultaneously monitored. Note that the crusted samples, Fig 3A, were initially crushed to get a small (nearly 20 mg) piece and was quickly placed inside the instrument.

The atmosphere in the sample chamber was purged with nitrogen to avoid oxidation or other undesired reactions. Once the weighing balance stabilized, the temperature of the chamber was increased at a rate of 5˚C/minute. The maximum temperature of the sample chamber was set to 540˚C in order to avoid any physical change in the glass spheres. The mass and temperature data were recorded every 0.1 seconds. One of the experiments was conducted with unused (dry) 0.13 mm diameter glass spheres to check for the accuracy of the measurements in case of dead weight. These experiments were repeated to get consistent trends.

Fig 6 shows the variation of the percentage mass loss, '$m_p$' (primary vertical axis) versus the chamber temperature for the crusted sample. Note that the maximum average surface temperature in evaporation experiments was about 60˚C (see S5 Fig in S1 File). Cooling-led condensation increases the trapped water content in the crusted sample. This extra condensed water evaporated rather easily. A peak in the mass rate curve (secondary vertical axis) for the crusted sample is seen at ~70˚C. Note that such a strong peak was not seen (data not included here) in case of the dry sample. The mass loss at temperatures higher than 100˚C is observed possibly due to enhanced potential energy of trapped water thanks to the sharp menisci. The other mass rate peak, corresponding to 310˚C, is due to the evaporation of adsorbed water; this peak was seen in both the samples. Nearly 1% of the mass loss is seen occurring until 100˚C majority of which would have been the condensate water. Out of the total mass loss of 2.5%, we can

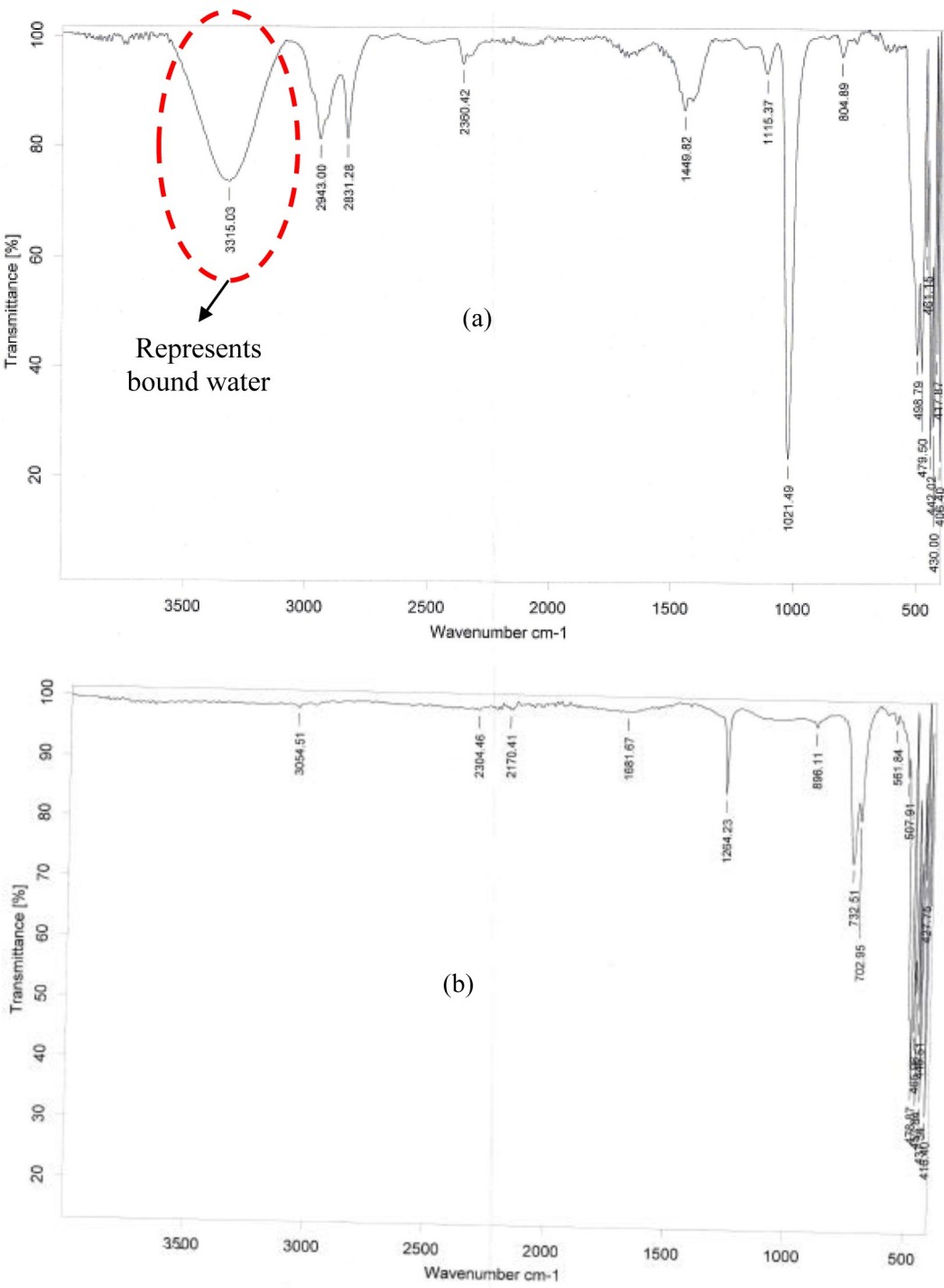

**Fig 5. FTIR spectrum showing transmittance values versus the wavenumber of two samples.** A strong peak at ~3300 cm$^{-1}$ in (a) indicates the presence of water in the crust. Such a peak was absent (b) in case of completely dry (unused) glass beads.

thus say that 1.5% of water was trapped originally in the crusted sample when the sample was not cooled; samples would automatically cool down once the IR heater is switched off. It is interesting that such tiny water content leads to an enormous increase in the overall strength

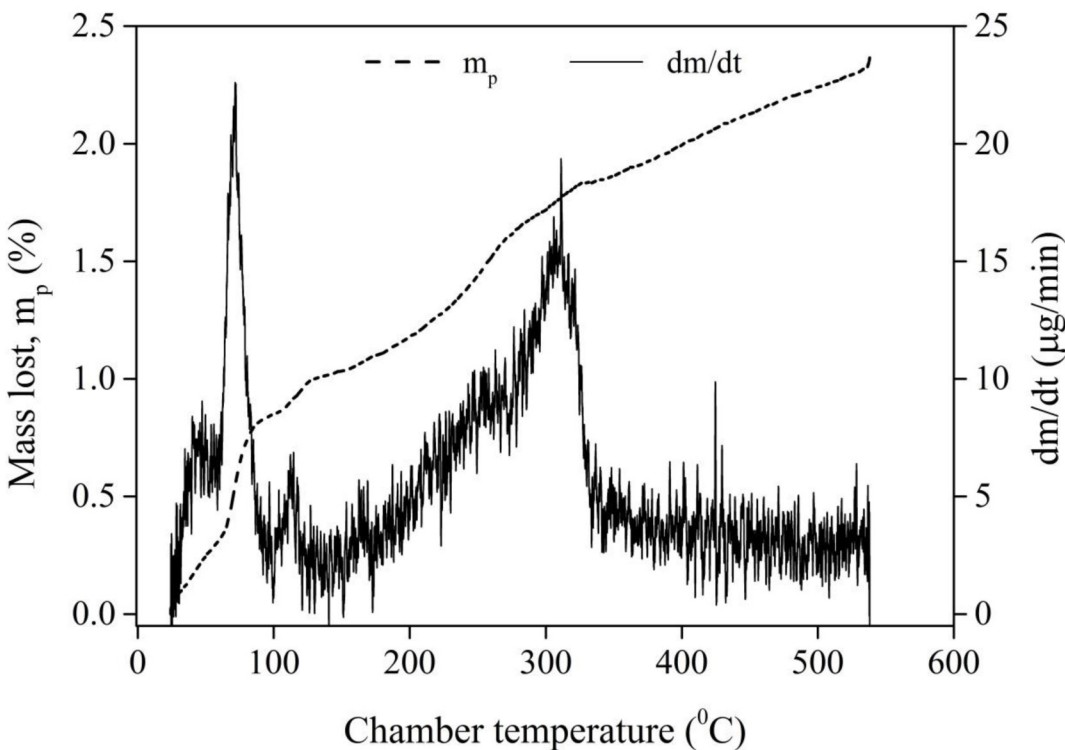

**Fig 6. Variations, with the TGA/DTA chamber temperature (horizontal axis), of the percentage (primary vertical axis) and rate of (secondary vertical axis) mass lost from the crusted sample.** The results with the unused sample are not included here.

of the crusts. We now investigate a key property of the crusts viz. crust thickness, for which we propose a unique strategy for this purpose in the next section.

## A unique method to determine crust thickness

Water gets trapped between the spheres either when it is leaving (due to preferential migration) a particular region or is evaporating. These regions can be distinguished, as in S9 Fig in S1 File, [25] if a coloured solution is used. This method works in a variety of porous systems as well [49, 50]. A few researchers have successfully used this method to determine the stage of evaporation. A small (larger than a two-sphere-system and smaller than the large scale system) scale experiment was designed to see the colour patterns in an evaporating porous medium. This experiment was performed with 0.13 mm diameter glass beads and coloured (using orange colour fluorescein dye) water in a small Teflon box. The diameter of the Teflon box was 3.2 cm and the height of the sample was 1.4 cm. Initially, the porous medium appears green throughout (Fig 7A) due to the presence of the fluorescein particles in the solution phase. Capillary film(s) brings water, and fluorescein dye, from deeper regions of the porous medium to its top where water evaporates leaving the fluorescein dye deposited on the beads' surfaces. Since fluorescein dye particles are orange, the distributed deposited dye appears orange (Fig 7B). The crust formed in this experiment was broken and seen (Fig 7B) as a thin layer, 1–2 mm (10–20 layers), consisting of nearly all the fluorescein dye used in the experiment. Crust bottom, inverted pieces in Fig 7B, and non-crusted regions (below the crust) both appear white, the true colour of the glass beads.

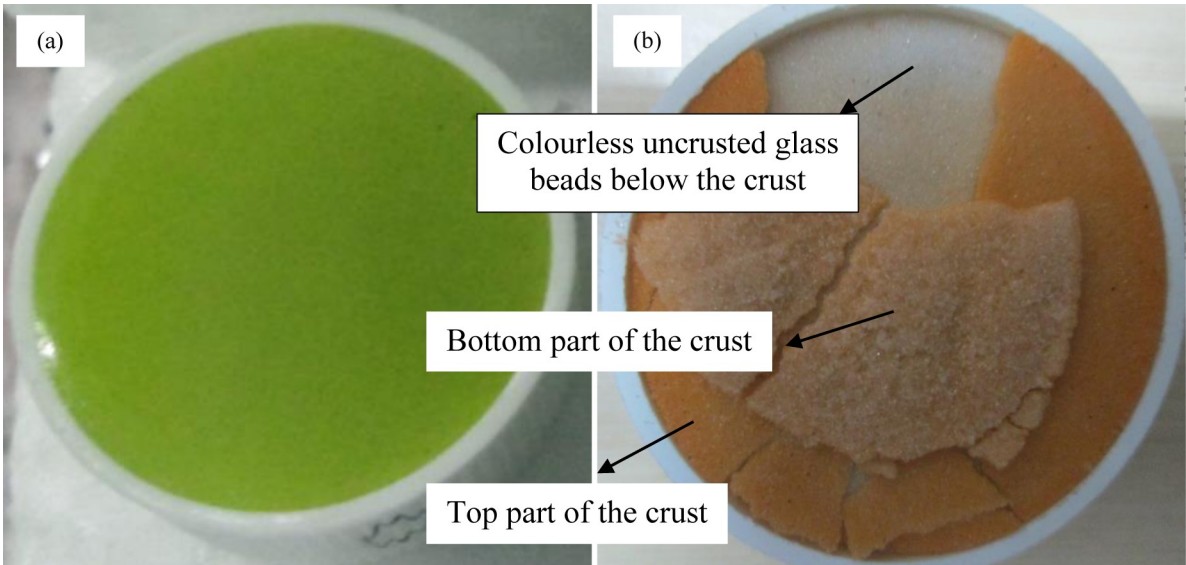

**Fig 7. Experiment with fluorescein dye and 0.13 mm diameter (hydrophilic) glass beads in a small Teflon container.** The porous medium is green throughout initially (a). The crusted thin upper layer is clearly seen in (b) at the end of the experiment. The lower regions of the porous medium do not show any significant deposition.

### Factors affecting the crust thickness

Deposited fluorescein dye in Fig 7B also shows that during stage 1 almost all the evaporation must occur within a thin layer near the exposed end of the porous medium. Since the particles would keep depositing (owing to drying) within these layers it is obvious that these layers form the crust. We now investigate the dependence of the crust thickness on various controlled parameters such as the particle size and the incident heat load (linked to the porous medium surface temperature). For studying the effect of varying heat flux, one set of experiments were conducted in small Teflon boxes, having a removable bottom, consisting of 0.13 mm diameter GB as seen in Fig 8A. The difference in samples' distance to the IR heater ensured different incident heat fluxes (this is governed by the varying view factors for different samples). Fig 8B shows the porous media top surfaces (of the four samples) after the experiment. The uncrusted particles, seen in Fig 8C were easily removed after the experiment leaving only the crusted layer attached to the container wall (S5 Video). This was required since the crusts in the previous experiments were needed to be broken (Fig 3A, Fig 7B) for their removal.

Table 1 shows a list of different types of experiments performed for this study. The saturated and unsaturated bulk densities of the samples across all the experiments varied within a small range 1.51±0.08 and 1.20±0.10 g/cc respectively. The average porosity of the samples was 36.5% with a narrow variation of ±1.5%. Crust thickness was calculated based on two different methods: (1) mass measurement of the crust and (2) fluorescein dye deposits. The average crust thickness [cm] was estimated, following a series of steps. These are,

$$\text{Average sample height [cm]} = \frac{\text{Total saturated mass of the sample [g]}}{\text{Saturated bulk density} \left[\frac{\text{g}}{\text{cm}^3}\right] * \text{Cross} - \text{sectional area [cm}^2]} \quad (2)$$

Here, the total saturated mass of the sample [g] is the addition of the masses of the glass beads

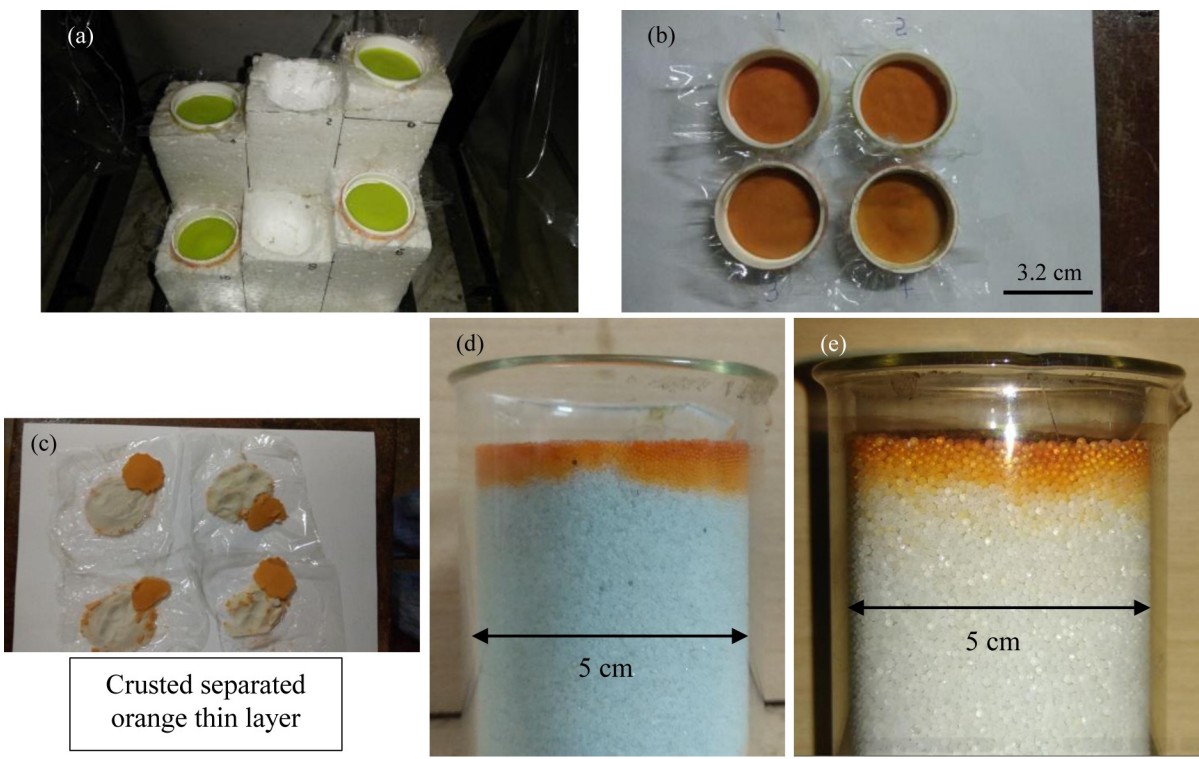

**Fig 8. Snapshots showing the condition of samples of crusts for different incident heat fluxes.** Different (hydrophilic) glass bead sizes were used for this study as well. The initial setup in (a) ensures different heating loads, for the four small containers (consisting of DI water, fluorescein dye, and 0.13 mm diameter hydrophilic glass beads), as the distances from its surface to the IR heater was different. Image (b) showing the conditions at the end of the experiment. The image (c) shows the removed hardened upper crusted layers for the different heat load experiments. The end conditions are seen for hydrophilic 0.45 mm diameter glass beads (d) and 0.78 mm diameter glass beads (e).

**Table 1. Experimental parameters in the present study.** Experiments with sample number '1–6' and '8' were conducted in small Teflon boxes, '9' in a medium-sized acrylic container, and '7' and '10–15' in large glass beakers. Hard layer thickness for the corresponding experiments is also mentioned. Average surface temperatures in stage 1 of evaporation are mentioned as a guide. The particles were hydrophilic in all the 15 cases mentioned here.

| Sample no. | Glass beads diameter (mm) | Incident heat flux (W/m²) | Column Height (mm) | Avg. surf. temp. in stage 1 (°C) | Crust / deposited dye thickness (mm) | No. of layers |
|---|---|---|---|---|---|---|
| 1 | 0.13 | 1400 | 9.7 | 44.5 | 1.8 | 17 |
| 2 | 0.13 | 1200 | 9.9 | 42.5 | 2.3 | 21 |
| 3 | 0.13 | 1000 | 9.9 | 40.0 | 2.5 | 24 |
| 4 | 0.13 | 700 | 9.7 | 36.0 | 2.2 | 21 |
| 5 | 0.13 | 500 | 9.7 | 32.5 | 2.9 | 25 |
| 6 | 0.13 | 0 | 9.7 | 25.5 | 8.8 | 84 |
| 7 | 0.13 | 0 | 62.9 | 25.5 | 12.6 | 120 |
| 8 | 0.45 | 1400 | 9.4 | 44.5 | 3.8 | 9 |
| 9 | 0.45 | 1200 | 18.5 | 42.5 | 4.7 | 11 |
| 10 | 0.45 | 0 | 64.3 | 25.5 | 32.2 | 69 |
| 11 | 0.78 | 2000 | 87.2 | 50.5 | 7.6 | 16 |
| 12 | 0.78 | 1000 | 87.3 | 40.0 | 10.7 | 21 |
| 13 | 0.78 | 500 | 86.8 | 32.5 | 17.5 | 29 |
| 14 | 0.78 | 250 | 84.9 | 29.5 | 26.3 | 38 |
| 15 | 0.78 | 0 | 53.6 | 25.5 | 37.8 | 51 |

[g] and the saturated mass of water [g].

$$\text{Saturated bulk density } [\text{g/cm}^3] = \frac{\text{Glass beads mass } [\text{g}]}{\text{Saturated water volume } [\text{cm}^3] + (\text{Glass beads mass } [\text{g}]/2.5]} \quad (3)$$

The value 2.5 in Eq (3) represents the density [g/cm$^3$] of a single glass bead.

$$\text{Average crust thickness } [\text{cm}] = \frac{\text{Crust mass } [\text{g}]}{\text{Crust density } [\text{g/cm}^3] * \text{Cross sectional area } [\text{cm}^2]} \quad (4)$$

Next, we estimate the unsaturated bulk density i.e. density of the cluster of glass beads in the absence of water.

$$\text{Unsaturated bulk density } [\text{g/cm}^3] = \frac{\text{Glass beads mass } [\text{g}]}{\text{Average sample height } [\text{cm}] * \text{Cross} - \text{sectional area } [\text{cm}^2]} \quad (5)$$

The estimated unsaturated bulk density is assumed to be the same as the crust density, which is valid for a large number of particles and layers in the crust. However, even for a smaller number of layers, the deviation from the true value (from a larger number of layers) is small (2%). Note that while estimating the unsaturated bulk density, small traces of trapped water and fluorescein dye in the crust was ignored since their contribution is negligible. The experimental parameters for the current study used to estimate the crust thickness and number of layers in it are seen in Table 1. A few experiments (sample numbers '8' and '9') were conducted with 0.45 mm diameter glass beads in different containers. Experiments with sample numbers '6, 7, 10, and 15' were conducted without external heating. Small container experiments were not performed for 0.78 mm diameter glass beads.

Average crust thickness was nearly 2 mm in (the majority of the) experiments with external heating in small Teflon boxes (see Table 1); irrespective of particle size and heat flux; the difference in crust thicknesses across the experiments is minute. The crust was very weak in experiments without the external heating and could not be removed cleanly. For these experiments deposited fluorescein dye layer thickness hold more meaning. Crust thicknesses for these non-heating cases are much higher than their heating counterparts. In the non-heating case, the deposited dye thickness is seen increasing with the increasing glass bead size. With external heating nearly same deposited thicknesses (~5mm), see Fig 8D and 8E, were observed for 0.45 mm and 0.78 mm diameter glass beads; these experiments are not mentioned in Table 1. Fig 9 shows the relation between the incident heat flux ($q''$) and the obtained hard layer for different particle sizes. Two major conclusions can be drawn (a) the crust thickness increases at lower $q''$ and (b) larger beads give thicker crust for a fixed $q''$. The experimental data is fit using a power law as seen in Fig 9. The exponent of crust thickness (in 'mm') and $q''$ (in 'W/m$^2$') curve is about -0.30 for all the three glass bead sizes investigated. Interestingly, evaporation rate in stage 1 of evaporation was found to vary nearly linearly with $q''$ [51] which can be obtained using a simple surface energy budget [25,26,46]; discussing this relation here is not in the interest of present study. In a completely different system, where mud-peels' thickness was theoretically obtained [45], the exponent of mud-peel thickness and the evaporation rate was -0.67. Note that for those experiments where no external heating was done, we calculated $q''$ directly from (hotter) ambient temperature and the wet bulb temperature.

## Conclusions

Similar to caking, we report the formation of crust (hard layer) during evaporation from different types of porous media. Strength-wise these crusts were harder than the sandcastles and

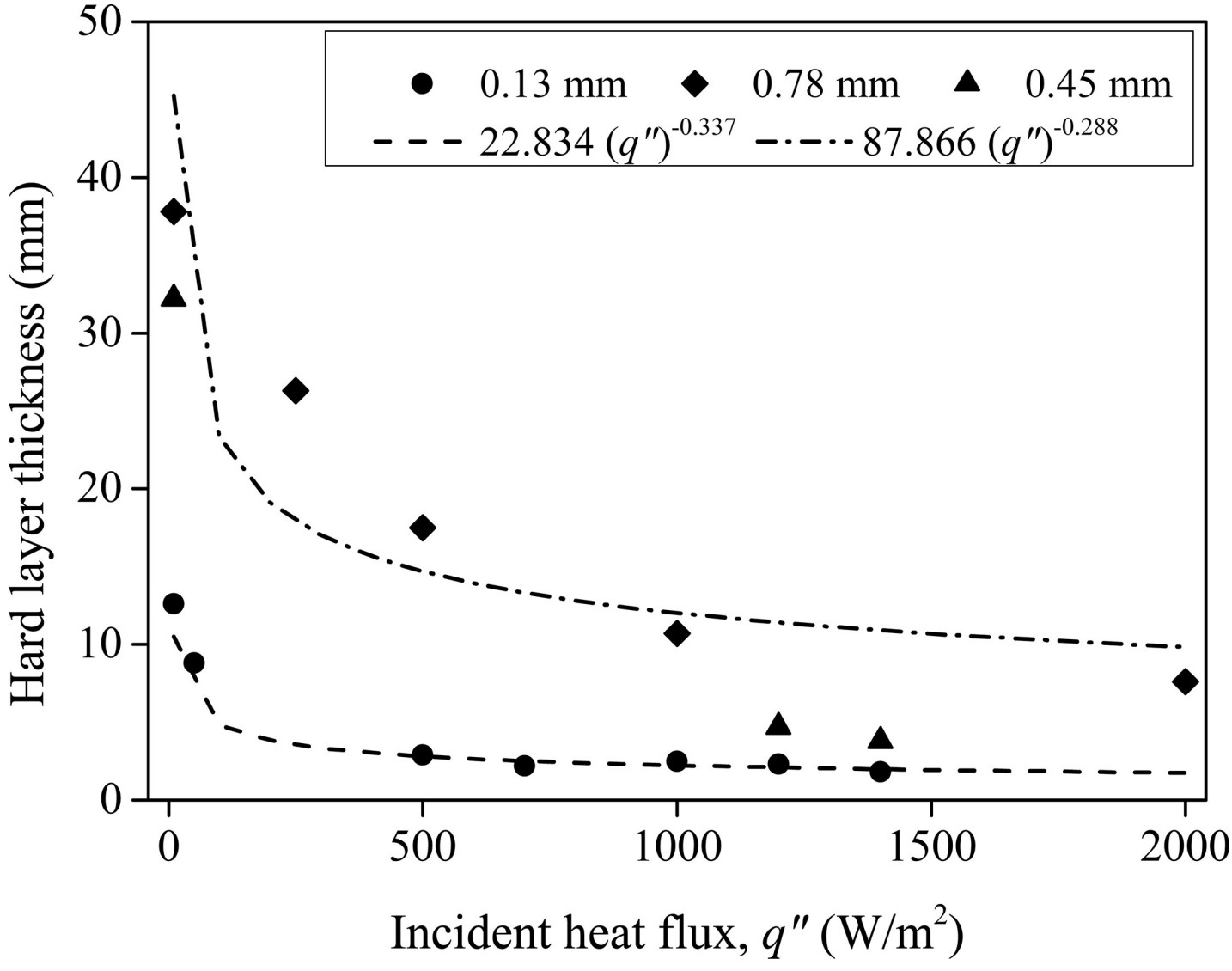

**Fig 9. Variation of the thickness of the near-surface hard layer formed in different experiments as a function of the particle sizes and the incident heat fluxes.** The data shown here is only for the hydrophilic particles.

were similar to mud-peels except, in the present study, they formed without salt (commonly known as 'leaching'). The present investigation thus focussed on the reasons, apart from leaching, behind the crust formation. The formation of the crust can be due to any of the following factors: (1) surface tension, (2) electrostatic force, (3) mechanical locking, (4) van der Walls forces, and (5) hydrogen bonding; any combinations of these five factors would strengthen the crust. However, in case of the sizes of the glass beads studied here, the surface tension seems to play the major role. Factors, such as the effects of particle size, heat flux, particle size variation, and the hydrophobicity of the porous medium (see methods section in S1 File), influencing the crust strength and its thickness were also investigated.

Weak crust (in glass beads) with acetone compared to water indicates that along with the hydrogen bonding between beads surface and water, the magnitude of the surface tension too played a key role in strengthening the crust. The dominant contributor is surface tension between the trapped water content and the beads' surface. Microscopic and SEM images

(showing a remarkably detailed view of the liquid bridge) clearly showed the presence of water trapped in the contacts of the beads. Presence of water in crusted samples was also confirmed using FTIR and TGA.

The strength of the crust in non-heating cases was found to be much weaker than with the heating cases. Crust strength was found to decrease drastically with increasing particle sizes. For particle sizes larger than 1.5mm crust either was very weak or did not exist even though liquid bridges did exist. We found the crust to be limited to a few layers near the top of the porous medium consisting of glass beads and confirmation was provided by deposited fluorescein dye layer thickness. The exponent of the crust thickness and the incident heat load (or the evaporation rate in stage 1) was obtained to be about -0.30; we do not have a theoretical basis for this behaviour. However, in the case of a natural sand experiment, the full column was found crusted and its strength was incredibly high. We expect all the factors to contribute considerably in making the crust tougher in this case. Shrinkage leading to detachment of sand particles from container wall was observed in this case, unlike the glass beads case where particles were in contact with the container walls and the liquid was trapped between them. A major missing point in the current investigation is the determination of crust strength and its hardness. Due to the unavailability of such a method, the crusts could not be tested for their strength and the information in the text was a completely hands-on experience, while breaking the crusts. Attempts will be made, in future, to analyze the crust strength in a quantitative way.

## Supporting information

**S1 File. Data, along with the associated videos, required for reproducing the results of this study are available at https://figshare.com/articles/Data_-_Crust_related_experiments/7430897.**
(DOCX)

**S1 Video.**
(AVI)

**S2 Video.**
(AVI)

**S3 Video.**
(AVI)

**S4 Video.**
(AVI)

**S5 Video.**
(AVI)

## Acknowledgments

We thank Prof. K.R.Y. Simha (Mechanical Engineering) and Prof. T.N. Guru Row (SSCU) of IISc for their valuable suggestions. We also thank Chemical engineering and Organic chemistry departments (IISc) for utilizing their experimental facilities. We are thankful to Mr Gautam Revankar who took the SEM images.

## Author Contributions

**Conceptualization:** Navneet Kumar, Jaywant H. Arakeri, Musuvathi S. Bobji.

**Data curation:** Navneet Kumar.

**Formal analysis:** Navneet Kumar.

**Funding acquisition:** Jaywant H. Arakeri, Musuvathi S. Bobji.

**Investigation:** Navneet Kumar, Jaywant H. Arakeri, Musuvathi S. Bobji.

**Methodology:** Navneet Kumar, Jaywant H. Arakeri, Musuvathi S. Bobji.

**Project administration:** Navneet Kumar, Jaywant H. Arakeri, Musuvathi S. Bobji.

**Resources:** Navneet Kumar.

**Software:** Navneet Kumar.

**Supervision:** Navneet Kumar, Jaywant H. Arakeri, Musuvathi S. Bobji.

**Validation:** Navneet Kumar, Musuvathi S. Bobji.

**Visualization:** Navneet Kumar.

**Writing – original draft:** Navneet Kumar.

**Writing – review & editing:** Navneet Kumar, Jaywant H. Arakeri, Musuvathi S. Bobji.

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
