## [Decision Letter · Decision Letter 0]

8 Jan 2020

PONE-D-19-34425

Formation of a hard surface layer during drying of a heated porous media

PLOS ONE

Dear Dr. Kumar,

Thank you for submitting your manuscript to PLOS ONE. After careful consideration, we feel that it has merit but does not fully meet PLOS ONE’s publication criteria as it currently stands. Therefore, we invite you to submit a revised version of the manuscript that addresses the points raised during the review process.

We would appreciate receiving your revised manuscript by Feb 22 2020 11:59PM. To enhance the reproducibility of your results, we recommend that if applicable you deposit your laboratory protocols in protocols.io, where a protocol can be assigned its own identifier (DOI) such that it can be cited independently in the future. For instructions see: http://journals.plos.org/plosone/s/submission-guidelines#loc-laboratory-protocols

We look forward to receiving your revised manuscript.

Kind regards,

Jie Zheng, Ph.D

Academic Editor

PLOS ONE

Journal Requirements:

Reviewers' comments:

Reviewer's Responses to Questions

**Comments to the Author**

1. Is the manuscript technically sound, and do the data support the conclusions?

Reviewer #1: Partly

Reviewer #2: Partly

2. Has the statistical analysis been performed appropriately and rigorously? 

Reviewer #1: N/A

Reviewer #2: Yes

3. Have the authors made all data underlying the findings in their manuscript fully available?

Reviewer #1: Yes

Reviewer #2: No

4. Is the manuscript presented in an intelligible fashion and written in standard English?

Reviewer #1: No

Reviewer #2: Yes

5. Review Comments to the Author

Reviewer #1: This paper reports the crust formation in heated porous media during water evaporation and concludes that the remaining water, as liquid bridge, is responsible for the formation and strength of the crusts. The liquid bridge shown in microscopic pictures are impressive. However, the organization of the manuscript should be carefully revised so I suggest that major revision is needed before this manuscript is considered publishable on PLOS One. Line numbers are not seen in this manuscript. Comments are listed below:

Abstract

1. The author should use past tense when describing the results of this study.

2. “Amount of the trapped water was ~1.5% (this is ~10 times higher than in the samples with caking), which was confirmed qualitatively using SEM images”.

Please rewrite this sentence.

Introduction

Page 1: How does “addition of a small amount of water” change soil strength, increase or reduce?

Page 2: Description of Stage 2 and 3 is missing.

Page 2: “textual layered”. I think a noun is missing here.

Page 3: “soil” was not seen in Materials and Methods, did the author mean “sand”?

Materials and Methods

1. The source (manufacturer and country of manufacturing) and grade/purity of each material should be described in this section.

2. FTIR, TGA and SEM procedures should be described here rather than in the later Results and Discussion section.

3. What were the containers used and what are the purposes?

4. What was the (volume) ratio of isooctane: FOTS?

Results and Discussion

1. Page 5: What was the porosity of the glass beads? Did hydrophobic and hydrophilic glass beads differ in porosity?

2. Page 6: The “bouncing stainless steel ball method” can be moved to Supporting Information.

3. Page 7: Regarding Figure 4, suggest adding a control picture (without liquid bridges) to compare.

4. Some data like FTIR and TGA in Supporting Information are even more important than those pictures in Figure 3, 5 and 6. There are no restrictions on number of figures so the author should consider including all important figures in the main text.

5. Page 11: please check the unit in equation (3).

6. Page 11: Table 1, how did the author get “no. of layers”?

7. Hydrophilic glass beads vs. hydrophobic glass beads: the author should note which glass beads were used in each experiment.

Conclusions

The conclusions were too long (655 words). Please shorten this section and only summarize the most important findings/conclusions here.

Reviewer #2: In this manuscript, the authors discuss the surface hardening process in a medium consisting of sub-millimeter spheres. Through control experiments with varied particle size, particle surface wettability, liquid medium and characterization with FTIR, TGA, SEM, etc, they determine that the capillary bridge plays a major role in the hardening phenomenon. In general, the manuscript is well written in academic format, the data is relatively solid and can support their conclusions, thus it can be published in Plos One if authors can address following issues after minor revisions:

1. As this paper conclude that the liquid bridge, or the capillary bridge, is the dominant contributor to the surface hardening, then the surface water wettability should be a major factor in the hardening result. Authors are using ‘hydrophobic’ spheres for most of their experiments. Could the authors provide some characterizations, such as contact angle, of the surface wettability before and after the hydrophobic treatment? The hydrophobic surface should have a water contact angle higher than 90 degrees. Maybe after the treatment, even though the contact angle increase, it is still less than 90 degrees and cannot be considered a ‘hydrophobic’ surface. Also, In Figure S11, the authors discuss a scheme for hydrophilic surface but not a hydrophobic surface.

2. This reviewer assumes that comparing to strength, the hardness may be a more important parameter describing the hardening process. In page 6, section-time of hardening-bouncing stainless steel ball method, the authors use the dropping ball methods to describe the hardening process, do authors have enlarged surface image after the ball ‘indentation’? Have authors tried any standard hardness measurement to characterize or these indentation images to calculate the hardness?

3. On page 7, first sentence, ‘these clumps could not be held between the fingers as they fell apart’ is not a very scientific expression, please rephrase.

4. Figure 3, please provide high-resolution images and clearly label the images, such as marking important features in the image, add scale bar, etc. Right now it's very unreadable.

5. Figure 4b, please add scale bar, Figure 4c, d, please enlarge the scale bar

6. Figure 5b, is the center brown region the crusted thin upper layer? If so, please mark carefully.

7. Figure 6, again, please label clearly on all subfigures, and please provide a scale bar for d and e.

6. PLOS authors have the option to publish the peer review history of their article (what does this mean?). If published, this will include your full peer review and any attached files.

Reviewer #1: No

Reviewer #2: No

---

## [Author Response · Author response to Decision Letter 0]

11 Feb 2020

We thank the Editor, the editorial board, and both the referees for entertaining our manuscript. Please see attached point-by-point response to the concerns raised by the referees. Thanks again.

---

## [Editor Report · Decision Letter 1]

13 Feb 2020

Formation of a hard surface layer during drying of a heated porous media

PONE-D-19-34425R1

Dear Dr. Kumar,

We are pleased to inform you that your manuscript has been judged scientifically suitable for publication and will be formally accepted for publication once it complies with all outstanding technical requirements.

With kind regards,

Jie Zheng, Ph.D

Academic Editor

PLOS ONE
---

## [Editor Report · Acceptance letter]

14 Feb 2020

PONE-D-19-34425R1 

Formation of a hard surface layer during drying of a heated porous media 

Dear Dr. Kumar:

I am pleased to inform you that your manuscript has been deemed suitable for publication in PLOS ONE. Congratulations! Your manuscript is now with our production department. 

With kind regards,

on behalf of

Dr. Jie Zheng 

Academic Editor

PLOS ONE